# Harnessing the Potential of PLGA Nanoparticles for Enhanced Bone Regeneration

**DOI:** 10.3390/pharmaceutics16020273

**Published:** 2024-02-15

**Authors:** Mozan Hassan, Hiba Atiyah Abdelnabi, Sahar Mohsin

**Affiliations:** Department of Anatomy, College of Medicine and Health Sciences, United Arab Emirates University, Al Ain P.O. Box 15551, United Arab Emirates

**Keywords:** nanoparticles, PLGA, bone regeneration, biomaterial, biocompatibility, biodegradability, bioactive molecules, drug delivery

## Abstract

Recently, nanotechnologies have become increasingly prominent in the field of bone tissue engineering (BTE), offering substantial potential to advance the field forward. These advancements manifest in two primary ways: the localized application of nanoengineered materials to enhance bone regeneration and their use as nanovehicles for delivering bioactive compounds. Despite significant progress in the development of bone substitutes over the past few decades, it is worth noting that the quest to identify the optimal biomaterial for bone regeneration remains a subject of intense debate. Ever since its initial discovery, poly(lactic-co-glycolic acid) (PLGA) has found widespread use in BTE due to its favorable biocompatibility and customizable biodegradability. This review provides an overview of contemporary advancements in the development of bone regeneration materials using PLGA polymers. The review covers some of the properties of PLGA, with a special focus on modifications of these properties towards bone regeneration. Furthermore, we delve into the techniques for synthesizing PLGA nanoparticles (NPs), the diverse forms in which these NPs can be fabricated, and the bioactive molecules that exhibit therapeutic potential for promoting bone regeneration. Additionally, we addressed some of the current concerns regarding the safety of PLGA NPs and PLGA-based products available on the market. Finally, we briefly discussed some of the current challenges and proposed some strategies to functionally enhance the fabrication of PLGA NPs towards BTE. We envisage that the utilization of PLGA NP holds significant potential as a potent tool in advancing therapies for intractable bone diseases.

## 1. Introduction

The urge for functional bone substitutes is rapidly growing nowadays; this can be attributed to the demographic aging phenomenon and the escalating number of bone grafting surgeries [1]. Currently, a plethora of assorted techniques exist to enhance bone regeneration procedures, including autogenous bone graft, which is often regarded as the gold standard treatment for large bone defects [2]. Nevertheless, the use of autogenous grafts has been limited due to a restricted quantity, a significant risk of donor morbidity, prolonged hospitalization, and elevated associated expenses. Allografts, while an alternative, are not without challenges, including potential immune reactions and the risk of transmitting infectious diseases [3].

The natural bone extracellular matrix (ECM) is composed of a hierarchical arrangement of composite materials in which nano-hydroxyapatite (nHA) crystals are disseminated within aligned bundles of collagen fibers [4]. Bones have a limited self-healing capacity when it comes to critical-size defects; in order to tackle this challenge, the concept of bone tissue engineering (BTE) has been introduced. BTE employs smart biomaterials to construct grafts that mimic natural bone structure and the ECM. This can be accomplished through a cooperative strategy that integrates a supportive scaffold biomaterial, acting as a platform to transport nutrients and facilitate interactions between cells and signaling molecules, ultimately promoting bone regeneration [5]. Biomaterials can be categorized into natural polymers, including substances like collagen, gelatin, silk fibroin, and chitosan; synthetic polymers, which encompass materials like polylactic acid (PLA), polyglycolic acid (PGA), polylactic-co-glycolic acid (PLGA), and polycaprolactone (PCL); ceramics, such as hydroxyapatite (HA) and β-tricalcium phosphate (β-TCP); bioactive glasses (BGs); and metals and composites, which combine two or more of the previously mentioned materials [6].

BTE strategies strive to produce a customized bone framework that closely aligns with the structure, bioactivity, and mechanical properties of natural tissue. The scaffold should be capable of providing a suitable environment for cell adhesion, proliferation, and differentiation [7] and possess both osteoconductive and osteoinductive characteristics for optimal performance [8]. When a substance is deemed biocompatible, it signifies that it seamlessly integrates with its surrounding environment, is not recognized as foreign, and consequently does not trigger any negative reactions. Bioactive material systems possess the remarkable capability to establish a robust bond with adjacent tissues through the activation of biological responses within cells. This process, in turn, promotes the direct formation of tissue on the material’s surface, ultimately augmenting the strength of the interface. Consequently, materials with osteogenic properties possess an inherent ability to facilitate the process of bone regeneration [9].

The bone is a highly dynamic organ and is constantly remodeling. Osteoblasts and osteoclasts work in tandem to facilitate the remodeling process of bones. Osteoblasts play a crucial role in the creation of new bone tissue, while osteoclasts are responsible for the removal of old bone tissue [10]. Bone remodeling is highly orchestrated by many signals and pathways to regulate the balance between bone formation and resorption [11]. Over the past years, many studies have been trying to mimic these signals and pathways by adding bolstering components to bone grafts, such as growth factors, stem cells, drugs, and proteins, to assist and enhance bone regeneration. Although the procedures to simulate bone ECM are still challenging in empirical research and the resulting grafts are not utterly identical to native tissue, prominent progress has been made in manipulating and modifying biomaterials. Nanotechnology has recently emerged as a promising field with the potential to significantly thrust ahead the field of BTE since cells interact with tissues at the nanometer scale [12].

## 2. Nanostructured Materials for Enhanced Bone Regeneration

Nanomaterials can be described as substances wherein their constituent parts possess at least one dimension smaller than 100 nm. Nanostructured materials such as nanofibers, nanotubes, and nanowires exhibit one dimension at the nanoscale, while the other dimension can extend beyond the micrometer scale in length [13]. These materials find extensive applications in medical and diagnostic fields. Nanofibers, characterized by a highly porous architecture that resembles the ECM, allow cells and nutrient accommodation and, moreover, enable the integration of growth factors and biofunctionalization to promote tissue regeneration [14]. Nanotubes exhibit tubular configurations at the nanoscale, offering various possibilities across diverse fields such as electronics, materials science, therapeutics, and diagnostics. Carbon nanotubes are particularly promising in areas such as bone regeneration as well as drug and gene delivery [15]. Nanowires, with their slender form at the nanoscale, find applications in electronics, sensing, and other advanced technologies [16].

Nanoparticles (NPs) are widely used in the biomedical and pharmaceutical sectors; they can replicate the structure and nanoscale features of bone while accurately reproducing important biochemical elements [12,17]. NPs offer a promising avenue for advancing bone regeneration techniques by providing targeted drug delivery, enhancing biocompatibility, promoting osteoconductivity, and enabling improved imaging and diagnostics. As research in nanotechnology and regenerative medicine progresses, the role of nanoparticles in bone regeneration is likely to expand further [18]. The synthesis process of NPs is amenable to manipulation and precise control of their size, morphology, and surface properties. This customization can result in enhanced cellular uptake and more effective interaction with the host’s immune and progenitor cells at the nanoscale level, leading to improved outcomes in bone regeneration [19]. In addition, the reduction in material size to the nanoscale significantly increases the surface area, surface roughness, and ratio of surface area to volume, leading to superior physicochemical properties [20]. NPs also exert influence on cell signaling, proliferation, and viability. Beyond supporting cellular functions, NPs can also control the behavior of osteoblasts, affecting their function, proliferation, differentiation, and migration [21]. Liang et al. have shown that HA NPs (55 nm) can promote osteoblastic differentiation and bone formation in rats with an expanded sagittal suture during expansion [22]. Another study by Huang et al. demonstrated that magnetic Fe_3_O_4_ NPs enhance osteogenic cell adhesion and differentiation in vitro by up-regulating the TGFβ-Smad pathway while simultaneously facilitating bone formation in rabbit femoral bone injury in vivo [23]. NPs were also able to up-regulate osteogenic-related genes and proteins and stimulate the production of vascular endothelial growth factor (VEGF), promoting angiogenesis in the cranial defect model [24].

### Nanoparticles Classifications

Generally, NPs are often categorized into three groups based on their composition: organic, inorganic, and carbon-based. Table 1 presents an overview of various NP categories, along with their respective merits and demerits.

In BTE, NPs can be incorporated into bone scaffolds to act as fillers and provide mechanical support, or they can be employed as carriers for delivering bioactive molecules that stimulate bone regeneration [30]. Recently, special attention has been directed toward polymeric nanoparticles (PNPs) due to their many features, such as biocompatibility, biodegradability, water solubility, and lack of immunogenicity [31]. In addition, PNPs are much cheaper than gold or silver NPs, and their morphology can be easily tailored as needed [32]. Moreover, the inner shell of PNPs is stabilized by hydrogen bonds and hydrophobic interactions, facilitating bioactive molecule encapsulation and protection and concurrently enhancing the drug’s solubility [33]. Although PNPs are readily available and frequently used, an optimal drug delivery system capable of transporting bioactive molecules to a specific target within bone remains a challenge. PLGA is one of the most effectively used biodegradable polymers; it undergoes hydrolysis and breaks down into naturally occurring metabolite monomers, namely lactic acid (LA) and glycolic acid (GA). These monomers are already present in the body and can be easily metabolized through the Krebs cycle [34]. This unique property has made PLGA an appealing and safe choice for drug delivery and biomaterial applications, as it minimizes systemic toxicity concerns.

This review provides an overview of some PLGA properties and recent advancements in PLGA-based NPs in systemic or localized strategies for targeting bones, particularly with a special focus on their synthesis techniques and drug-loading techniques. These insights may open up new possibilities for delivering drugs using PLGA nanocarriers to precisely address bone-related conditions. Furthermore, the review explores the efficacy and safety of PLGA NPs, as well as their application forms in scaffold constructs such as electrospinning, 3D printing, nanofillers, gas foaming, and leaching. Additionally, the review touches upon some commercially available PLGA-based NP products that have successfully made it from benchside to clinical use. Finally, we are going to highlight some of the current challenges and future perspectives regarding PLGA NPs and their use in BTE applications.

## 3. PLGA Nanoparticles and Their Properties

PLGA is a linear copolymer of different ratios of LA and GA monomers (see Figure 1). PLGA has gained significant attention in recent years due to its many advantageous properties, including excellent biocompatibility, favorable biodegradability, controllable mechanical characteristics, and endorsement by regulatory bodies like the US Food and Drug Administration (FDA) and the European Medicines Quality Agency (EMA) [35]. Delivery systems based on PLGA have shown great potential in the treatment of bone disorders, and currently, a wide range of pharmaceutical formulations, including microspheres, hydrogels, NPs, and more, are available on the market or undergoing clinical trials [36].

PLGA is a versatile copolymer and has been extensively studied as a potential carrier for various molecules, including drugs, proteins, DNA, RNA, and peptides [37,38,39]. To create an effective controlled drug delivery system using PLGA, it is crucial to understand its physical and chemical properties. These physical properties can be influenced by factors like the nanoparticle synthesis method, the molecular weight of PLGA, and the incorporation of active ingredients, surfactants, and other additives [40]. Accordingly, the drug release characteristics of PLGA can be optimized by adjusting its composition, molecular weight (Mw), and chemical structure [41]. Typically, when a growth factor or molecule encapsulated in NPs is aimed at the bone regeneration process, NPs become nested within a secondary system, such as hydrogels or sponge scaffolds. These secondary systems also influence the release pattern of these molecules from NPs [42]. Eventually, the synthesis, encapsulation, and surface modification processes will be integral to developing systems aimed at achieving controlled release.

### 3.1. Modulating PLGA Properties for Enhanced Bone Regeneration

#### 3.1.1. PLGA Physicochemical Properties

PLGA is usually synthesized through ring-opening co-polymerization, where monomers are linked by ester bonds. Another synthesis method is polycondensation of LA and GA, which is usually used to obtain low-molecular-weight PLGA [43]. The ratio of poly (LA) to poly (GA) can be adjusted to create various forms of PLGA (e.g., 80/20, 75/25, 60/40, and 50/50), providing flexibility in tailoring its characteristics [44]. PLGA combines properties of both LA (rigidity, hydrophobicity, and gradual degradation) and GA (pliability, reduced hydrophobicity, and faster degradation); hence, the choice of LA/GA ratio and the molecular weight of the polymer significantly impact PLGA’s hydrophobicity, crystalline structure, mechanical properties, size, and biodegradation rate [45]. Increasing the proportion of GA leads to higher hydrophilicity and greater degradability [46], while a greater LA proportion exhibits reduced hydrophilicity, resulting in decreased water absorption and extended degradation time [45].

The crystallinity of PLGA ranges from fully amorphous to fully crystalline, determined by its block structure and molar ratio. PLGA copolymers synthesized by combining poly (D, L-lactide) and poly(glycolide) exhibit an amorphous structure, while those derived from poly(L-lactide) and poly(glycolide) display crystalline properties. Additionally, it is worth noting that PLGA containing less than 70% poly(glycolide) is also amorphous [47]. Improving the crystallinity of PLGA can be employed as a means to concurrently alter its degradation characteristics [48]. PLGA exhibits a glass transition temperature between 40 and 60 °C and can be dissolved using various solvents. The solubility of PLGA is influenced by its composition, allowing it to dissolve in a diverse array of solvents [49]. For bone scaffolds, the ratio of LA to GA should be tailored according to injured bone mechanical properties, whereas higher LA concentrations are needed for more mechanically stable scaffolds.

#### 3.1.2. Biodegradation

PLGA undergoes degradation through the cleavage of ester bonds and subsequent dissolution [50]. A three-stage degradation model was suggested by Linbo et al. [51], namely the quasi-stable stage, the loss of strength stage, and the disruption of scaffold stage (see Figure 2). The first stage involves a reduction in the scaffold dimension and an increase in the mechanical strength, while the weight remains the same. This is followed by a significant drop in mechanical strength due to molecular weight loss. Later, the final stage is marked by significant degradation, weight loss, dimension reduction, pH reduction caused by the release of acidic degradation products (LA and GA), increased fragility, and changes in pore morphology until the scaffolds eventually disintegrate. The exact timing and characteristics of these processes can vary depending on the scaffold material and composition.

PLGA proportions frequently employed in biomedical research include 50:50, 65:35, 75:25, and 85:15. Among these, PLGA 50:50 is typically the choice for drug delivery systems [52]. Typically, PLGA 50:50 degrades the fastest, followed by PLGA 65:35, which is attributed to the higher hydrophilicity causing preferential degradation of the GA component. Subsequently, PLGA 65:35 degrades faster than PLGA 75:25, and PLGA 75:25 degrades faster than PLGA 85:15 [39]. Another crucial factor in customizing PLGA properties is the length of its chain. This is because its physical strength and degradation rate are significantly influenced by its molecular weight. Increasing PLGA molecular weight from 10–20 to 100 kDa will result in variation in the degradation rates from a few weeks to several months [53].

Recent research has shown that the degradation of PLGA can be harnessed to achieve controlled drug release. Figure 3 summarizes some factors that can affect the drug release mechanism of the PLGA polymer. Lin et al. [54] successfully created a precise core-shell microsphere delivery system using 50:50 PLGA. This system exhibited excellent control over the release of Mg^2+^ ions, leading to improved growth and differentiation of osteogenic cells. In a rat model, 75% of the newly formed bone tissue was adequately mineralized compared to a control group, and the regenerated tissue displayed an impressive retention of 96% of the original bone tissue’s mechanical strength [54]. Another study displayed a sustained release of transforming growth factor beta1 (TGF-β1) by encapsulating it in PLGA (50:50) NPs that are embedded within collagen scaffolds. The system was able to mimic the gradual release of TGF-β1 typically seen in native human bone ECM [55]. Additionally, incorporating PLGA (50:50) NP into a chitosan/BG scaffold has significantly improved the scaffold’s mechanical strength, making it similar to cancellous bone and enabling controlled drug release for a long-term frame [56]. A study by Koopaei et al. [57] found that encapsulating the anticancer drug docetaxel in pegylated PLGA NPs led to a reduction in tumor size and growth in mouse models while minimizing the drug’s adverse side effects. In vitro experiments demonstrated an initial burst release of the drug, followed by a sustained release pattern, and the docetaxel-encapsulated NPs exhibited stronger cytotoxic effects on ovarian cancer cells compared to free drugs [57]. Controlling drug release from PLGA also involves considering the drug concentration. Higher drug concentrations can lead to increased water absorption, which in turn promotes the formation of pores, ultimately accelerating the release of the drug [58].

#### 3.1.3. Mechanical Strength

PLGA holds great potential in BTE; however, a key challenge involves precisely adjusting its mechanical properties to match those of the surrounding tissue. The Young’s modulus of pure PLGA is 2 GPa [59,60], whereas the Young’s modulus of human bones varies due to factors such as anatomical location, measurement techniques, measuring conditions (wet or dry), and test direction [61]. Some literature indicates Young’s modulus ranging between 10–20 GPa and 23–26 GPa for human cancellous and cortical bones, respectively [62]. In contrast, other studies proposed a lower range of 0.05–0.5 GPa for cancellous bone and 7–30 GPa for cortical bone [63]. Additionally, research using ultrasonic and mechanical techniques suggests Young’s modulus of 10.4–14.8 GPa for cancellous bone and 18.6–20.7 GPa for cortical bone [64]. Nanoindentation techniques revealed Young’s modulus of 15–19.4 GPa for trabecular bone and 16.6–25.7 GPa for cortical bone [65].

In order to increase PLGA stiffness and mechanical properties, many studies have explored using it in composites such as TiO_2_, HA, calcium phosphate, and BG. Fiedler et al. reinforced PLGA’s mechanical stiffness by adding TiO_2_ NPs. They were able to imitate the Young’s modulus of different bone tissues by adding different fractions of TiO_2_, in which increasing TiO_2_ resulted in a higher Young’s modulus, indicating the potential to fine-tune material properties for specific applications in bone-related research [66]. Also, Park et al. showed that PLGA-grafted HA composites were able to increase scaffold tensile strength more than double while also enhancing biocompatibility [67]. Another study revealed that interference screws composed of PLGA/β-TCP composites exhibited negligible mass reduction over a period of six months, suggesting that the material retained its mechanical integrity and shape with time [68]. Furthermore, the 3D-printed PLGA/TCP/Mg scaffold showed enhanced mechanical properties when implanted in rabbit ulnar bone defects; the scaffold was also able to promote osteogenesis and angiogenesis [69]. Moreover, Magri and colleagues demonstrated that PLGA/BG composites exhibited superior in vitro cell proliferation and enhanced in vivo bone formation when compared to BG/collagen composites [70]. Additionally, to improve PLGA’s mechanical properties, Duan et al. [71] proposed the use of a bilayer PLGA scaffold in the treatment of osteochondral defects using a rabbit animal model. After 24 weeks of implantation, Young’s modulus of the newly formed tissue was approximately half that of normal cartilage, and the physiological characteristics closely resembled native tissue [71]. To sum up, creating customized scaffolds based on composition and mechanical properties is a practical approach for meeting the unique needs of the target tissue to be regenerated.

#### 3.1.4. Particle Size and Morphology

Nanoscale carriers present numerous benefits compared to larger particles. They exhibit enhanced versatility by remaining stable in colloidal solutions and facilitating even distribution. Furthermore, their diminutive size enhances the bioavailability of encapsulated molecules, and their substantial surface area-to-volume ratio enables easy surface modifications. Additionally, they can penetrate the cell more efficiently for targeted drug delivery [42]. A slight alteration in the average particle size can significantly impact the properties of NPs, ultimately affecting their effectiveness in delivering therapeutic molecules to cells. Sahin et al. [72] demonstrated that larger NPs (230.8 nm) exhibited greater encapsulation efficiency when contrasted with their smaller counterparts (157.9 nm). Nevertheless, smaller NPs were more efficient in intracellular drug delivery [72]. Large NPs are eliminated rapidly by either phagocytic cells or kidneys; controlling particle size can be achieved by carefully choosing the fabrication technique. The double emulsion and spray drying techniques typically result in the generation of relatively large NPs, often exceeding 300 nm in size. In contrast, nanoprecipitation has been employed to create smaller NPs, typically ranging from 100 to 200 nm in size [73]. Huang and Zhang’s study revealed that the size of PLGA NPs is greatly dependent on parameters related to the coefficient of solvent in water, such as polymer concentration, organic solvent, temperature, and ionic strength. A high diffusion coefficient results in smaller NPs, whereas decreasing it can increase the overall particle size and distribution [74].

Recently, many studies have utilized microfluidic systems to control the size of PLGA NPs. This technology enables precise control of liquids in small volumes, making it ideal for creating micro-scale reactions for droplet formation. The key advantage of utilizing this technology lies in its ability to finely tune preparation parameters, making it an appealing choice for enhancing encapsulation formulations [75]. Bao et al. [76] were able to develop size-tunable PLGA NPs using a microfluidic device without the need to modify the polymer’s molecular weight, concentration, or composition. By employing a high flow rate, they successfully produced small NPs (less than 200 nm). Moreover, after loading the chemotherapeutic drug paclitaxel into these NPs, smaller-sized NPs (52 nm) demonstrated enhanced in vitro anti-tumor activity and cellular uptake compared to larger NPs [76]. Another study demonstrated that utilizing a microfluidic system is superior to the traditional manual mixing method for controlling the size of NPs. This approach not only enhances NP size control but also preserves all the desirable characteristics of PLGA NPs [77].

The shape of NPs can also facilitate their cellular uptake, with rods showing the highest uptake, followed by spheres, cylinders, and cubes. Non-spherical NPs have advantages in terms of biological performance, including prolonged circulation in the bloodstream, reduced removal by immune cells, and passive accumulation within cells [78]. Modifying the diameter and shape of NPs can control both their accumulation extent and depth of penetration within cells, whereas larger NPs (>100 nm) struggle to move beyond blood vessels and get trapped in the ECM between cells. In contrast, the smallest NPs (<20 nm) can penetrate deep into tissues but are not retained beyond 24 h [79].

### 3.2. Surface Modifications

Numerous challenges confront the effectiveness of PLGA NPs. These include rapid clearance from the bloodstream, reducing their lifespan, and their limited ability to be recognized by diseased tissues for targeted therapy. Moreover, PLGA NPs and cell membranes both have a negative surface charge; this similarity in charges increases their vulnerability to phagocytosis and hampers their uptake through endocytosis [78]. As a result, many studies suggested surface modifications to improve PLGA NP’s efficacy; the main modifications and their importance are shown in Figure 4.

#### 3.2.1. PEGylation

PLGA PEGylation, which involves adding polyethylene glycol (PEG) to PLGA, is one of the commonly used modifications to enhance the stability and improve the biocompatibility of PLGA-based drug delivery systems [78]. PEG has an active hydroxyl terminal, enabling it to be coupled with vast active drug molecules [80]. PEG-PLGA, loaded with osteogenic factors, can stimulate osteogenesis. Yan et al. [81] found that PLGA-PEG-PLGA loaded with simvastatin can maintain sustained drug release and augment mineralization and osteogenic gene expression. Whereas in vivo, it showed enhanced bone formation in rat animal models [81]. Another study revealed that bioceramic porous scaffolds incorporating simvastatin-loaded PLGA-PEG NPs exhibited dual functionality, promoting both osteoinductivity and osteoconductivity. Consequently, it improved the healing of calvarial bone defects in a rat model [82]. Han et al. [83] suggested a hybrid injectable hydrogel delivery system containing chitosan microspheres loaded with stem-derived exosomes and PLGA-PEG-PLGA NPs loaded with VEGF. The system was able to enhance angiogenesis and osteogenic differentiation in vitro, while in vivo it promoted bone formation in calvarial bone defects [83].

#### 3.2.2. Surfactants

The addition of surfactants is another modification for PLGA NPs to improve their colloidal stability. Surfactants work by reducing surface tension at the interfaces between different components within the system, resulting in improved solubility, uniform particle size, and better dispersion [84]. PLGA NPs commonly coated with polyvinyl alcohol (PVA) surfactant, a study by Istikharoh et al. showed that scaffold composed of nHA/PLGA/PVA exhibited exceptional characteristics, including optimal porosity, biodegradability, and enhanced surface roughness, making it an ideal biomaterial for treatment of orthopedic injuries [85]. A different investigation revealed that PLGA-PVA NPs have the capability to extend the release duration of bone morphogenetic protein (BMP), enabling a sequential discharge of BMP-2 followed by BMP-7, mimicking natural tissue behavior. This ultimately enhances osteogenic differentiation while leaving the mechanical properties of the scaffold unaffected [86]. Various surfactants, such as poloxamers, polysorbates, sodium cholate, and vitamin E, are employed in conjunction with PLGA NPs. The specific surfactant type and its concentration can influence the stability of these NPs, modulate the release characteristics, and impact the efficiency of encapsulation. Consequently, they play a pivotal role in governing the uptake of these NPs by cells [87].

#### 3.2.3. Phospholipids

PLGA NPs have a hydrophobic surface, which makes them vulnerable to removal by immune cells. To address this issue, numerous studies have explored modifying the surface of PLGA NPs with phospholipids to improve their stability and evasion of the immune system [88]. Li and colleagues demonstrated that the type and concentration of phospholipids can impact physicochemical properties, drug release profiles, and cellular uptake by macrophages [89]. Synthetic lipids such as 1,2-dioleoyl-3-(trimethylammonium) propane (DOTAP) offer the benefit of being easily processed and tailored when employed in the surface modification of PLGA NPs. Furthermore, the integration of natural cell membrane lipids, which are present in erythrocytes, leukocytes, platelets, and stem cells, imparts unique cell-mimicking properties to the surface of these particles [78]. Natural membranes possess the ability to evade immune detection, enabling immune escape. Additionally, these membranes are equipped with membrane proteins that facilitate specific cell binding, thus enabling active targeting [90].

#### 3.2.4. Surface Ligands

A targeted drug delivery system involves transporting a bioactive substance or drug exclusively to a specific tissue or organ. This technique offers several benefits, such as precise tissue targeting, improved bioavailability, and minimum side effects [91]. The approach involves the incorporation of targeting ligands into PLGA NPs capable of interacting with a molecule highly expressed in the specific tissue of interest [92]. Recent advances have explored the use of nuclear factor-kappa B (NF-κB) decoy, which is an oligonucleotide ligand with an NF-κB binding site. This ligand exhibits the ability to entrap the NF-κB transcription factor, thereby effectively inhibiting its pro-inflammatory activity [93]. Huang et al. showed that PLGA NPs modulated with NF-κB decoy can inhibit inflammation of the extracted tooth socket, which is triggered by exaggerated osteoclast activity, and also improve alveolar bone healing [94]. Another study displayed that curcumin-loaded PLGA NPs conjugated with folic acid are effective in targeting cancer cells expressing folate receptors, while in vivo they resulted in tumor size reduction in a mouse model [95]. A further study used annexin A2 (AnxA2) antibody-conjugated curcumin-loaded PLGA NPs against cancer cells expressing AnxA2 surface antigen [96]. Bone-targeting ligands for drug delivery were extensively reviewed by Xu et al. [97]. These include targeting osteoclastogenesis through receptor activator for nuclear factor-κB ligand (RANKL), targeting bone metabolism through sclerostin, targeting calcium/phosphorus metabolism through type 1 parathyroid hormone receptor (PTH1R), targeting membrane expression receptors through colony-stimulating factor 1 receptor (CSF1R), integrins, and sphingosine 1 phosphate receptor (S1PR), targeting cellular crosstalk by semaphorins, and targeting gene expression such as Sp7, Runx2, and tumor suppressor genes [97]. Other surface-decorating ligands that can be used for targeted therapy include tumor necrosis factor receptor 1 (TNFR1) on macrophages, intercellular adhesion molecule 1 (ICAM1) on the endothelium, and vascular cell adhesion molecule 1 (VCAM1) for leukocytes [98]. The physical characteristics of particles, specifically their size and ligand density, play a crucial role in determining their ability to effectively target specific tissues, potentially limiting their overall therapeutic efficacy [99].

## 4. PLGA Nanoparticles: Therapeutic Uses

To achieve the desired therapeutic efficacy, a nanocarrier must satisfy three crucial prerequisites. Firstly, it should securely encapsulate the active ingredient and release it efficiently upon reaching the intended target. Secondly, it must maintain a low profile within the bloodstream to evade detection by the reticuloendothelial system. Lastly, the nanocarrier should possess the capability to infiltrate specific cells at the precise location where therapeutic action is required [100].

This review focuses on the use of PLGA in bone repair and regeneration; however, researchers in different medical fields have used PLGA as a drug delivery system to deliver various pharmaceutical agents. PLGA nanosystems can be used to load small drug molecules such as chemotherapeutics, antimicrobials, antioxidants, etc., and macromolecules such as proteins, growth factors, and genes. Listed in Table 2 are some examples of studied PLGA nanosystems. In the following sections, PLGA NP synthesis techniques and their use in bone therapy are discussed in detail.

## 5. Techniques for PLGA-Based Nanostructure Preparation

Polymers can be fabricated in many different formulations, alone [120,121,122] or in conjunction with other polymers [119,123,124] or nanosystems [125,126,127] for their use as drug delivery systems in nanomedicine. The biocompatibility, lipophilicity, and gradual degradation properties of PLGA make it a suitable drug delivery system for sustained release purposes [113,128,129] and localized therapy [102,116,130]. PLGA can be formulated using different techniques to produce NPs [101], nanospheres [131], nanofibers [132], microspheres [133], and scaffolds [123]. These techniques include, but are not limited to, emulsions [134], nanoprecipitation [118], electrospray [107], salting out [119], electrospinning [135], etc. Figure 5 shows different techniques for the preparation of PLGA nanostructures, while Table 3 summarizes some of their advantages and disadvantages.

### 5.1. The Emulsion–Solvent Evaporation Method

PLGA can load/encapsulate both hydrophilic and lipophilic drugs using single [56,108] or double [116,131] emulsion methods. In single emulsions, the lipophilic active agent is mixed with PLGA in an organic phase, which is then added gradually to an aqueous phase while magnetic stirring to form an oil-in-water (O/W) emulsion. The organic phase is then evaporated to get the generated NPs [56]. As for hydrophilic drugs, a W/O/W double emulsion is needed to be encapsulated in PLGA [144,145]. Similarly, solid-in-oil-in-water emulsion (S/O/W) may be used to encapsulate drugs in their solid forms [117]. Marquette et al. used S/O/W to encapsulate anti-TNF alpha into PLGA microspheres [120]. The most commonly used aqueous solutions in the preparation of such emulsions contain surfactants (stabilizers) such as different percentages of PVA [103,111,145], poloxamer 188 [108], and vitamin E (TPGS) [56,118]. This method is one of the simplest methods to formulate PLGA NPs. The emulsion–solvent evaporation method may result in different particle types, such as NPs [110,111], microparticles [146], microspheres [120,133], and nanospheres [131]. PLGA molecular weight as well as its concentration, aqueous phase pH, stabilizer type and its concentration, homogenizer type, and speed are all important parameters in optimizing particle size, polydispersity index (PDI), particle surface charge (zeta potential), and encapsulation efficiency [116,145,147,148].

### 5.2. Nanoprecipitation Method

Likewise, PLGA NPs can be obtained from nanoprecipitation, also called the phase separation method [121]. As mentioned previously by Barichello et al. [149] and Govender et al. [150], the drug and PLGA are dissolved in a water-miscible organic phase, usually acetone, and then injected into an aqueous phase with a stabilizer, usually poloxamer 188 [114,115,139]. This method is typically used for the entrapment of lipophilic agents [112,118,139]. Hydrophilic drugs have low encapsulation efficiency [115,121]. PLGA and stabilizer concentrations have a great influence on the particles obtained [147,151].

### 5.3. Electrospinning Method

The electrospinning technique is a simple method of producing different sizes of uniform fibers with a 3D nanostructure that is similar to bone ECM. It has been frequently used in the fabrication of PLGA nanofibers that can be used as scaffolds for bone regeneration [129,132,140,152,153]. In this method, electrospinning equipment is required to extrude the electrospun solution through a spinneret toward a rotating drum under high voltages. Based on the arrangement of the spinneret, the electrospinning process can be categorized into needleless, needle-based, coaxial, and triaxial configurations, each producing systematically organized fibrous structures [154]. The electrospun solution is prepared by dissolving PLGA polymer in an organic solvent along with the drug under vigorous stirring for an elongated time until complete dissolution [132,152]. The distance between the needle tip and the rotating drum, the rotation speed of the drum, the flow rate and viscosity of the solution, voltage, and the needle tip diameter are important parameters that control the structure of the fabricated fibers [140,153]. Moreover, the type of solvent used and its physiochemical properties, especially surface tension, boiling point, dielectric constant, and viscosity, can highly influence nanofiber morphology, fiber diameter, drug encapsulation efficiency, in vitro release, and nanofiber mechanical properties [154].

Electrospinning is very promising in drug delivery systems. Yao et al. used electrospinning to coat calcium phosphate cement (CPC) with drug-loaded silk fibroin/PLGA nanofiber to overcome its weak biocompatibility and enhance its osteoinductivity [155]. For tissue regeneration and wound healing purposes, PLGA/gelatin nanofibers were loaded with ciprofloxacin and quercetin. These nanofibers were found to aid epithelization and collagen formation to enhance wound healing in vivo [156]. One of the main challenges associated with employing electrospun nanofibers in drug delivery systems lies in the need to achieve sustained drug release while avoiding initial bursts. A recent approach to addressing this challenge involves the utilization of coaxial electrospinning, where core-sheath nanofibers are fabricated. In this technique, a polymer nanofiber serves as the core and is enveloped by another polymer, enabling the controlled delivery of a specified amount of drug over a designated duration, tailored to the medical condition. This method also helps to avoid the potential toxicity linked to post-treatment procedures applied to electrospun fibers, such as crosslinking and chemical modifications [154].

### 5.4. 3D Printing Method

3D printing became one of the favored methods for scaffold formation due to its ability to tailor the product and prepare patient-specific and customized scaffolds in a cost- and time-saving manner [143,157]. Moreover, 3D printing has different techniques, such as but not limited to fused deposition modeling (FDM), extrusion-based bioprinting, and 3D low-temperature solvent-based printing technology, using different printing machines. FDM methods require high temperatures, so they are not the best choice for all materials [157]. However, polymers such as PCL, PLA, and PLGA are compatible with them. Babilottea et al. used FDM in their study to fabricate PLGA/HA scaffolds for bone regeneration, and in vitro results showed that the scaffold is safe without inflammation signs and enhances cell proliferation [158]. Low-temperature solvent printing is an alternative method that avoids high temperatures [157,159]. Moreover, 3D printing produces scaffolds with regular shapes, uniform porous architecture, and satisfactory mechanical strength that resemble the natural bone structure [159,160,161]. PLGA LA/GA ratio, polymer solution composition, viscosity, temperature, and the method and machine used in the printing procedure all affect the printed scaffold [143,157,158,160,161].

### 5.5. Other Methods

Furthermore, many other methods are used to fabricate PLGA polymers, such as the salting-out method, which produces spherical PLGA NPs [119], the melt-spinning method, in which the fibers are produced through heating to a high temperature that melts the polymer and then extruded through a high-speed spinning mesh [162], and the solvent coating/leaching method [123,163,164]. In the latter method, also known as solvent casting/particulate leaching technique, polymers are dissolved in organic solvent and then cast on salt porogen, and then the solvent is evaporated over a long time. An aqueous solution is added to the matrix, dissolving the salt-forming polymer films [165].

## 6. Fabrication Forms of PLGA Particles

Due to its biocompatibility, controllable degradability, ability to be formulated with other polymers, and ease of handling, PLGA is a very suitable polymer fabricated in various forms for bone treatment. PLGA has been prepared as NPs, nanospheres, microparticles, microspheres, micelles, as well as fibrous scaffolds.

### 6.1. PLGA as Nano- or Microparticles

Emulsion–solvent evaporation and nanoprecipitation methods are widely used to formulate round, spherical PLGA NPs and microspheres. The preparation of PLGA polymers as nano-/microparticles makes them suitable for intravascular and intramuscular injections [121,130]. Due to their hydrophobicity, PLGA particles are used as drug delivery systems to prolong the half-lives of the loaded drugs and control their release [116,138]. As with other nanosystems, PLGA NPs’ surfaces may be functionalized with bone tissue-targeting molecules such as poly-aspartic acid sequences [166], zoledronate [139], tetracycline [167], or alendronate [168]. Moreover, to enhance cell adhesion, proliferation, and osteogenic differentiation in vitro, PLGA nano-/microparticles can be encapsulated in other polymer scaffolds or hydrogels [137]. Collagen scaffolds are usually used to load PLGA particles [106,131,146,153]; for example, Wang et al. loaded the PLGA microspheres within the formed collagen/HA scaffold [169]. Others used chitosan [56,133] or PLLA/PLGA/PCL [170] as final scaffolds to encapsulate the PLGA particles.

### 6.2. PLGA Scaffolds

As we mentioned earlier, for bone-tissue-critical defects, systemic therapies are not enough to heal the bones. Therefore, artificial scaffolds should be implanted locally to allow new stem cell attachment and differentiation into osteoblasts in the injured bones [143,171,172]. PLGA can be fabricated in scaffolds having natural bone properties mainly by using electrospinning, 3D printing methods, or other methods that will result in a porous structured scaffold (see Figure 6). However, PLGA properties can be enhanced by adding other polymers, ceramics (inorganic components), or both to the PLGA materials. Likewise, PLGA/ASP-PEG scaffolds were fabricated by Pan et al. and Lin et al. teams by solvent casting/particulate leaching techniques [123,163]. Furthermore, inorganic ceramics (such as biphasic calcium phosphate BCP and micro-nano bioactive glass MNBG) are combined with PLGA to improve its mechanical strength, wettability, bioactivity, cellular adhesion, and proliferation, control its degradation rate, maintain pH levels upon PLGA degradation, and make it similar to biological ECM [49,132,139,152,157,160,170,171,173]. In addition, polymeric scaffolds can be loaded with other NPs and active agents for further improvement in drug release profile and scaffold cell function and to reduce systemic side effects [56,133,153,169].

## 7. PLGA-Loaded Bioactive Molecules for Bone Regeneration

As mentioned above, bone-critical defects are challenging to treat, and bones are not easily healed. Artificial scaffolds must simulate the ECM and permit cellular adhesion, proliferation, and differentiation on their surfaces. Hence, therapeutic agents like drugs, growth factors, peptides, DNA, and ions should be loaded on the scaffolds. PLGA scaffolds can be bioactivated with such agents to improve their function and accelerate bone regeneration (see Figure 7). Therapeutic agents may be added to the scaffolds in various ways, through physical attachment [174,175] or chemical modification/immobilization [135,164]. Different bioactive loads of PLGA scaffolds are discussed in the following section.

### 7.1. Peptides

#### 7.1.1. BMP-2

The most extensively studied protein in bone therapy is the bone morphogenetic protein BMP. BMP is known to have good osteoinductive, osteoconductive, osteoblast differentiation, and bone regeneration functions [176,177]. BMP-2 and BMP-7 for clinical use have been approved by the FDA. rhBMP-2 was encapsulated into PLGA NPs, resulting in a prolonged release profile and enhanced cell differentiation [138]. Using poly-dopamine (PDA) activation, BMP-2 was immobilized onto PLGA/HA scaffolds; this combination resulted in an additive effect on cell differentiation [162]. PLGA formulated with other polymer scaffolds proved a sustained release of rhBMP-2 and promoted bone regeneration in vivo, such as in rhBMP-2/PLGA-alginate/Collagen-HA scaffolds [146] and PLGA/HA/Chitosan/rhBMP-2 scaffolds [159]. Zhu and his team formulated BMP-2-encapsulated PLGA microspheres and then loaded them into PLLA/PLGA/PCL scaffolds; in vivo, results indicated active bone repair, an increment in bone mineral density (BMD), and upregulation of bone genes [170]. However, BMP-2 is expensive and unstable, so a less costly synthetic peptide may replace it, such as P24 [163].

The P24 peptide is a derivative peptide from BMP-2, which has a stable linear structure containing many phosphorylated serine and aspartic acid residues [178,179]. P24 has been studied for bone regeneration and bone defect repair purposes. A dextran and hydroxypropyl chitosan polysaccharide hydrogel containing PLGA/HA microspheres loaded with P24 was prepared and tested in vitro and in vivo for the treatment of bone defects. The composite hydrogel showed osteoinductive and osteoconductive abilities [136]. Similarly, a bilayered scaffold of P24 peptide loaded at the surface of a PLLA/PLGA/PCL nanofibrous scaffold using PDA and kartogenin-loaded hydrogel was prepared and tested for osteochondral repair. The results showed that the bilayered scaffold enhanced chondral and subchondral bone regeneration [180]. The 3D printing method was used with different research groups to fabricate P24-loaded PLGA scaffolds. Duel active agents of the disinfectant chlorhexidine and P24 were loaded into PLGA/TCP using graphene oxide and collagen; this scaffold proved to have antimicrobial properties along with osteogenic activity [142]. A multifunctional scaffold for the treatment and prevention of tumor recurrence was fabricated using the 3D printing method as well. PLGA/TCP scaffold loaded with black phosphorus (BP) nanosheet, doxorubicin (DOX), and P24 was printed as a hierarchical porous scaffold and showed excellent chemotherapeutic activity accompanied by bone regeneration ability in vitro and in vivo [141].

Specific bone genes/proteins that are usually studied are ALP, with its levels increasing in the early stages of osteogenic differentiation; Runx-2, which is important in the early stages of bone formation and responsible for other osteogenic gene transcription; collagen type 1 (Col1), which is expressed in the early stages and regulates bone remodeling; osteocalcin (OCN) and osteopontin (OPN), which are transcribed in the late stages of osteogenic differentiation; and OPN, which causes cell adhesion and increases mineralization [133,146,169,181].

#### 7.1.2. Other Proteins

VEGF is an essential cytokine for angiogenesis and bone development during bone regeneration [182]. However, its effect on bones is mainly visualized in combination with BMP-2. In vitro studies showed a synergistic effect on osteogenesis upon the combination of VEGF/BMP-2 with increased levels of ALP, Runx-2, OCN, and Col 1 [169,183]. Furthermore, the VEGF-functionalized PLGA/HA scaffold showed a controlled release profile, improved osteogenic differentiation, and higher levels of OCN, Runx-2, OPN, Col 1, and VEGF in vivo [181]. Moreover, the basic fibroblast growth factor (bFGF) was immobilized on PLGA/HA/graphene oxide in combination with BMP-2, which synergistically increased osteogenic differentiation and related gene expression (ALP, Runx-2, OPN) in vitro [132]. Bone marrow mesenchymal cells (BMSC) express high numbers of insulin receptors, and it has been shown that insulin induces cell proliferation and osteogenic differentiation through the elevation of ALP and mineralization [131]. In a very detailed study, Lee and his team fabricated a multifunctional PLGA composite containing magnesium hydroxide (MH), decellularized ECM, demineralized bone matrix, and polydeoxyribonucleotide (PDRN). They found that this composite not only has a synergistic effect on the upregulation of osteogenic and angiogenic-related genes but also has anti-inflammatory and immune-modulation roles [184].

### 7.2. Drugs

Local bone treatment has several advantages. Some bone defects cannot be treated without the implantation of drug-rich scaffolds. Drugs loaded into PLGA scaffolds may be used for bone repair, bone regeneration, and bone tumors. Cholecalciferol (vitamin D3) was incorporated into PLGA/HA NPs for bone regeneration purposes, and it proved its activity in vivo [130]. Simvastatin, which is a lipid-lowering medication, can increase the expression of osteogenic genes, resulting in osteoblast proliferation and differentiation. Simvastatin was encapsulated into PLGA microspheres that were further fabricated in chitosan/HA scaffolds to control its release and achieve synergistic bone formation activity [133]. Some non-steroidal anti-inflammatory drugs (NSAIDs) also have effects on bone regeneration; aspirin, for example, was formulated in PLGA NPs and then loaded into collagen nanofibers with curcumin. This scaffold showed satisfactory results in vitro by increasing ALP, Runx-2, and OCN, as well as cells completely occupying the defective area, replacing the scaffold without any inflammatory signs in vivo [153]. A natural active compound usually used in Chinese traditional therapy for osteoporosis, astragaloside (AS), was recently incorporated in mPEG-PLGA nanomicelles, and alendronate (AL) and tetracycline (TC) were used as targeting ligands toward bone tissues. AS/AL/mPEG-PLGA micelles improved the oral bioavailability of AS and its bone accumulation, resulting in enhanced bone mineral density and mechanical strength of osteoporotic bones in vivo [168]. AS/TC/mPEG-PLGA micelles alleviated the cytotoxicity of AS when administered IV as well as accelerated osteoporotic bone repair [167]. Polylevolysin (PLL) and fibronectin (FN) are part of the ECM. PLL is an amine-containing polymer that acts as a coating material for negatively charged cells by enhancing their electrostatic attraction, thus enhancing osteoblast adhesion and proliferation [185]. FN is the responsible polymer for the deposition and integrity of collagen in the ECM, and it is produced from osteoblasts along with type Ⅰ collagen [186]. Although their study has some limitations, Canciani and his team fabricated a PLGA/HA/dextran scaffold loaded with PLL and FN. An in vivo study showed that the PLGA scaffold has osteoinduction activity and increased bone regeneration after 6 months of implantation [175]. Since bone scaffolds are implanted into the bone-defected area, bacterial inflammation signs may appear. Ilhan et al. have prepared PLGA NPs for local delivery of clindamycin for alveolar bone regeneration, which have sustained release for up to 3 months upon a single injection [116].

Bone tumors can also be treated with PLGA scaffolds loaded with anticancer drugs. Doxorubicin (DOX) was entrapped into lamellar HA/PLGA scaffolds enwrapped with PDA for sustained DOX release; the scaffolds showed anti-tumor as well as osteogenesis activity [135]. Moreover, DOX was encapsulated in PLGA microspheres and then loaded into HA/collagen scaffolds to form post-surgery filling material that can inhibit tumor recurrence [106]. Additionally, zoledronate/PLGA/docetaxel NPs were prepared for targeted drug delivery systems for bone metastasis [139].

### 7.3. Ions

The most widely used inorganic component of PLGA is HA. HA, Ca_10_(PO_4_)_6_(OH)_2_, is the same as the natural physiological bone mineral composition. nHA has osteoinductive and osteoconductive functions; however, due to its brittleness and instability, it is usually combined with other osteogenic systems [174]. nHA is dispersed equally and uniformly on PLGA scaffolds, increasing its mechanical strength, hydrophilicity, mineralization capability, and osteoblast adhesion. PLGA/HA scaffolds have been tested in vitro and in vivo alone or in combination with other agents, such as other polymers and drugs. PLGA/HA/gelatin and PLGA/HA/collagen scaffolds proved to have enhanced osteogenic proliferation activity [140,143,152]. PLGA/HA microspheres showed accelerated bone mineralization activity along with enhanced osteoblast proliferation and differentiation in vivo [187]. PDA and polyethyleneimine (PEI) were used to chemically immobilize RGD peptides at the surface of PLGA/HA scaffolds [123,164]. PLGA/HA scaffold was also fabricated using the 3D printing method, and then the scaffold was soaked in gelatin solution to create a gel-filled PLGA/HA/gelatin scaffold [143].

Magnesium ions (Mg^2+^) promote osteogenesis and angiogenesis and inhibit osteoclast activity. Magnesium oxide (MgO) has been entrapped into PLGA/alginate microspheres that control the release of Mg^2+^, which causes increased levels of Col 1, ALP, OPN, and neuronal calcitonin gene-related polypeptide-α (CGRP) in vivo [54]. MgO was also combined with quercetin, which has anti-inflammatory, antiallergic, and anti-cancer activity, in PLGA scaffolds for bone repair [188]. Mg^2+^ can also maintain the environmental pH at normal levels after LA release upon the degradation of PLGA, protecting against inflammation progression. MNBG is an inorganic functional material that releases Ca and Si. MNBG can increase osteoblast proliferation, differentiation, and angiogenesis through its ability to increase the gene expression of osteogenic and angiogenic-related peptides (ALP, OPN, OCN, Runx-2, CD-31, and VEGF). MNBG can be incorporated into PLGA scaffolds alone or in combination with Mg [160,171,172]. Phosphorus ions can also increase the expression of osteogenic-related genes. Black phosphorus quantum dots (BPQDs) (phosphorus nanosheets) may be encapsulated into PLGA nanospheres and formulated into thermally induced hydrogels for targeted bone tumor therapy. BPQD proved to have anti-tumor and bone repair activity due to its high cell penetration ability and photothermal conversion efficiency [137,189]. Strontium-zinc ions were combined in a PLGA/HA composite scaffold to evaluate its compressive strength and ability to act as bone substitutes [50]. Owing to its mechanically favored properties, titanium dioxide (TiO_2_) has been conjugated with PLGA scaffolds to increase its mechanical strength. The PLGA/TiO_2_ scaffold prepared by the 3D printing method was tested in vitro for its osteogenic activity, and results showed enhanced cell proliferation, increased Ca deposition on the scaffold, increased protein adsorption, which enhanced cell adhesion, and increased ALP levels, indicating cellular differentiation [161].

## 8. Cytotoxicity and Safety Evaluation of PLGA Nanoparticles

PLGA NPs have garnered approval for numerous applications in the field of biomedicine due to their outstanding characteristics and remarkable adaptability. Nonetheless, safety concerns regarding the potential toxic effects of these particles were raised. Upon administration or implantation, the degradation byproducts of NPs may potentially accumulate in several organs, causing adverse immune responses or inflammation and accelerating bio-corona formation. The nature of this toxicity can range from acute to chronic, depending on the characteristics of NPs and the composition of surrounding biomolecules [190]. There is a suggestion that NP nanoscale size may lead to a greater exposure of molecules to the surface when compared to larger particles; this increased exposure can potentially result in a higher occurrence of oxidation reactions, hence the production of reactive oxygen species (ROS) [191]. On the other hand, the nano-size feature is important for cellular internalization, as previously mentioned. NPs of ≤ 100 nm are reported to exhibit better endocytosis, circulation half-life, and pharmacokinetic behavior [192,193,194,195].

Several studies have assessed and proven the safety of PLGA NPs. Semete et al. [196] evaluated PLGA NP cytotoxicity in vitro, and their results showed that the cell viability was >75%, which was higher than other types of NPs such as zinc oxide, ferrous oxide, and fumed silica of the same size. Furthermore, when they administered PLGA NPs orally to mice and analyzed the in vivo distribution over 7 days, they found that the majority of the NPs were accumulated in the liver, followed by the kidney and brain. Importantly, there were no signs of inflammation or tissue necrosis in these organs, indicating the potential safety of these NPs [196]. Another study assessed the influence of PEG-PLGA NPs on pregnant mice and found no discernible impact on the weight of either the mother or the developing fetuses, indicating their safety [197]. Kim et al. [198] investigated the impact of early embryonic exposure to PLGA NPs on fetal development and subsequent generations. Their research found that embryos exposed to PLGA NPs exhibited normal and healthy development without any observed genetic abnormalities or mutations [198]. Creemers et al. [199] conducted a phase I clinical trial, affirming the safety of PRECIOUS-01, an immunomodulatory nanomedicine based on PLGA. This innovative formulation co-encapsulates a tumor antigen (NY-ESO-1) and a T cell activator [199]. Another study proved the biological safety of juglone-loaded PLGA NPs in mice, which is a natural plant dye with known anti-tumor activity, and demonstrated their efficacy in suppressing the growth of melanoma cells [200]. Significant progress has been made in the development of PLGA-based smart nanomaterials that possess the ability to respond to specific stimuli in a controlled manner, leading to changes in their physiochemical and functional properties. However, their practical use in clinical applications faces challenges.

Although numerous studies suggest the safety of PLGA NPs, the absence of in vivo data raises concerns about their effectiveness and safety in human trials. To move forward with human clinical trials, it is essential to gather more in vivo data in humans to evaluate both the efficacy and potential toxicity of PLGA NPs. Only when safety and efficacy are thoroughly confirmed can a pharmaceutical formulation be considered successful.

## 9. Commercial Products Based on PLGA

The PLGA polymer has gained significant popularity in regenerative medicine, largely due to its FDA approval for clinical use. Recent years have seen considerable advancement in PLGA-based materials within the field of regenerative medicine. Currently, there are several commercially available PLGA-based products in various forms, such as membranes, sponges, powders, gels, and sutures, each with different ratios of LA to GA. These products exhibit a wide range of degradation times, spanning from a few weeks to one year [201]. Table 4 summarizes some commercially available PLGA-based products along with their clinical usage, advantages, and disadvantages.

Polyglactin 910, also known as Vicryl suture, is a synthetic absorbable suture sometimes coated with an antibiotic agent to prevent bacterial infection after surgical procedures (coated Vicryl Plus). It is made from a copolymer of glycolide and lactide, which are biodegradable and bioabsorbable polymers. The exact composition and ratio of glycolide to lactide can vary depending on the specific type of suture and its intended use. Polyglactin 910 sutures are designed to gradually break down in the body over time, making them suitable for internal sutures that do not need to be removed after a certain healing period [202]. OsteoScaf^TM^ scaffold, composed of a unique combination of PLGA and calcium phosphate, emerges as an innovative material for bone replacement. It holds significant potential as a viable option for preserving alveolar bone structure following tooth extraction [203].

Biosteon^®^ is a blend of calcium–HA osteoconductive particles within a PLLA matrix to enhance durability preservation, bone integration capability, and pH stabilization during the graft healing process [204]. Bilok^®^ is an innovative calcium composite technology used in interference screws for ligament restoration and suture anchors in rotator cuff repairs. This cutting-edge material is made through a confidential manufacturing process, ensuring even dispersion of β-TCP particles within the PLLA matrix. This integration enhances the structural integrity of the components. Bilok utilizes a PLLA matrix with a low molecular weight and low crystalline structure, resulting in faster degradation and increased hydrophilicity, ultimately improving performance characteristics [205].

ActivaScrew^TM^ Interference and Milagro Advance Interference Screw are devices designed for fixing tissues such as ligaments, tendons, or bone-tendon connections while ensuring proper immobilization or controlled mobilization. They are composed mainly of bioabsorbable PLGA and are primarily used in orthopedic surgeries involving the knee, shoulder, elbow, ankle, foot, and hand/wrist regions [206].

Biosure Regenesorb Interference Screw is a biocomposite made of β-TCP/PLGA/calcium sulfate with an open-architecture design to allow bone ingrowth. In vivo animal testing has shown that Regenesorb material is bioabsorbable and is replaced by bone; additionally, it remains mechanically stable for a minimum of 6 months before subsequently being absorbed and replaced by bone within 24 months [207]. PLGA-based orthopedic devices are in extensive use and exhibit suitable degradation times. Nevertheless, their clinical performance is still under debate, and uncertainties persist regarding the clinical significance of incorporating osteoconductive materials into bioresorbable screws.

**Table 4 pharmaceutics-16-00273-t004:** Commercially available PLGA-based products for clinical use.

Product Name	Composition	Clinical Usage	Advantages	Disadvantages	References
Polyglactin 910 (Vicryl suture)	Copolymer of glycolide and lactide	Internal suture	Low friction, easy to handle, and fast absorption	Can cause inflammation if it remains in the skin for more than 7 days, causing scar tissue or stitched sinuses	[202,208]
Coated Vicryl Plus	Copolymer of glycolide and lactide coated with an antibiotic agent	Surgical incision suture	Prevent bacterial infection at the surgical site	Low efficacy in oral, breast, and cardiac surgeries	[202,209]
OsteoScaf^TM^ scaffold	PLGA and calcium phosphate	Clot-retention device and osteoconductive support for bone growth	Preserve the alveolar bone structure following tooth extraction	low mechanical properties and local acidification of PLGA can lead to clinical failure	[203,210]
Biosteon interference screw	HA particles within a PLLA matrix	Reconstruction of anterior cruciate ligaments and suture anchors for rotator cuff repairs	Osteoconductive material and HA particles improve strength retention, bone-bonding potential, and pH buffering during graft healing	Differences in the resorption rates between PLGA and HA particles could induce potential complications	[204,211]
Bilok interference screws	β-TCP particles within a PLLA matrix	Ligament restoration and suture anchors in rotator cuff repairs	Enhances structural integrity, faster degradation, and increased hydrophilicity	Screw can fracture during insertion or after insertion	[205,212]
ActivaScrew^TM^ Interference screw	Proprietary blend of PLGA	Fixation of tissue, including a ligament, tendon to bone, or bone–tendon to bone	Easy guided insertion and high strength; after operation, screw dimensions slightly change, improving the screw’s fit and isoelasticity	- Cannot be used in early weight-bearing rehabilitation due to their elasticity - Additional casting is required to maintain reduction and alignment	[206,213]
Milagro Advance Interference Screw	70% PLGA and 30% β-TCP	Attachment of soft tissue grafts or bone-tendon-bone grafts to the tibia and/or femur during the cruciate ligament reconstruction procedure.	Rapid insertion, excellent fixation strength, and enhanced bone engagement	Marrow edema around bone tunnels was seen 3 months after the operation and reduced after 6 months	[214,215]
Biosure Regenesorb Interference Screw	β-TCP/PLGA/ calcium sulfate	Fixing ligaments, tendons, soft tissues, or bone-tendon-bone grafts in knee surgery	Open architecture allows bone ingrowth through the screw and attachment to the graft, increasing strength	Require a special surgical fixation technique	[207,216]

## 10. Current Challenges and Future Perspectives

PLGA nanomaterials are widely studied and utilized in various fields, particularly in drug delivery, tissue engineering, and diagnostics. However, their development was not without limitations. It is important to note that research in this field is ongoing, and herein we present some of the current challenges and potential future strategies to overcome these obstacles. One of the main challenges with PLGA NPs is the initial burst release of the encapsulated drug. This can be problematic when precise and sustained drug delivery is required, especially if the drug in use can result in adverse reactions if released in excessive amounts. Another challenge is the control of particle size. Achieving a consistent and narrow size distribution of PLGA NPs can be challenging, as it depends on various formulation parameters. Variability in size can affect drug release kinetics and cellular uptake. In this review, we mentioned some fabrication techniques that can be useful to manipulate particle size and uniformity. Furthermore, PLGA NPs are prone to aggregation, especially when exposed to biological fluids with high salt concentrations, thereby potentially impacting their stability and drug release profiles. To mitigate this concern, it is generally advisable to subject NPs to in vitro testing conditions that closely mimic physiological environments.

The biodegradation rate of PLGA exhibits variability, mainly influenced by the ratio of LA to GA. This variability poses challenges in engineering NPs with precise degradation profiles; thus, selecting an appropriate LA/GA ratio is crucial for fine-tuning PLGA’s degradation kinetics. An additional obstacle is the incompatibility of PLGA NPs with heat-sensitive compounds, especially when using an emulsion/solvent evaporation production technique, which exposes heat-sensitive drugs to elevated temperatures, risking their degradation. Moreover, the heat sensitivity of PLGA NPs makes them unsuitable for conventional sterilization methods like autoclaving. Surface modification of PLGA NPs for specific targeting or controlled release is complex and can affect stability, complicating their design and application. Collectively, these challenges impede the transition of PLGA NPs from laboratory-scale to large-scale production, making consistent quality and reproducibility difficult. However, researchers are actively addressing these limitations through innovations in PLGA NP formulation, surface modification, and optimization techniques.

Prospects for the future could include progress in drug loading and controlled release, achieved through diverse strategies like surface modification, co-encapsulation with other materials, and the use of specialized drug carriers. Additionally, methods like microfluidics and nanoprecipitation can be harnessed to attain superior control over both the size and distribution of PLGA NPs. Employing techniques like PEGylation and surface modification can extend circulation time and enable precise targeting of specific tissues or cells. Another valuable avenue for improving treatment outcomes is the adoption of combination therapy, wherein multiple drugs or therapeutic agents are encapsulated. Ultimately, the customization of PLGA NPs for personalized medicine, tailored to individual patients based on their genetic and physiological characteristics, may emerge as a prominent healthcare trend.

## Figures and Tables

**Figure 1 pharmaceutics-16-00273-f001:**
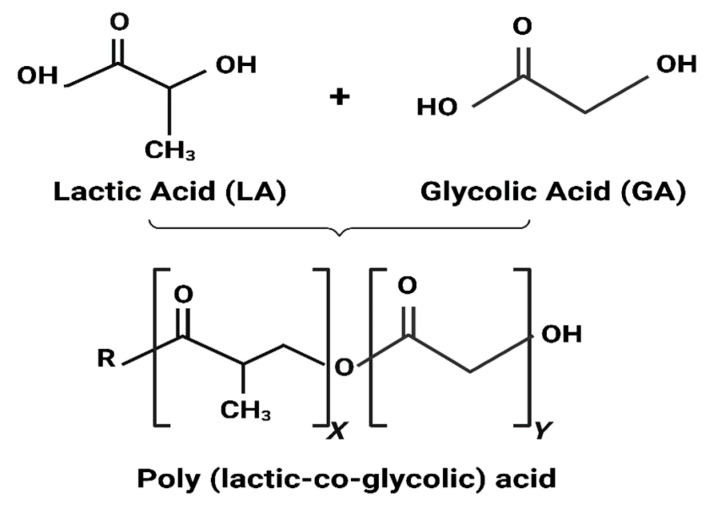
PLGA chemical structure.

**Figure 2 pharmaceutics-16-00273-f002:**
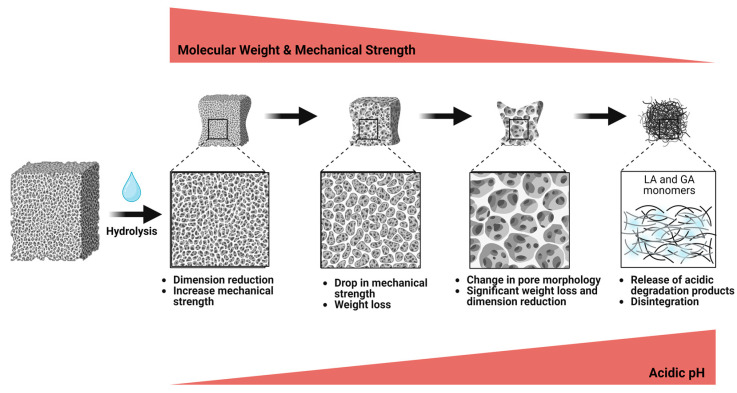
Biodegradation stages of the PLGA polymer.

**Figure 3 pharmaceutics-16-00273-f003:**
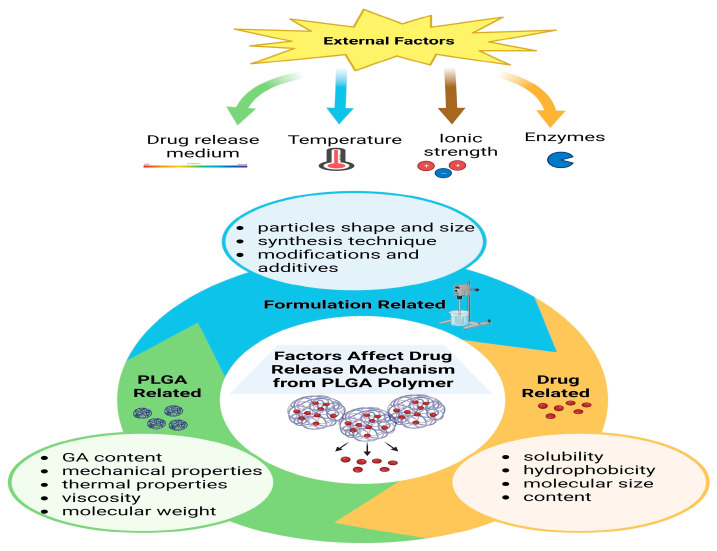
Factors that affect drug release from PLGA nanoparticles.

**Figure 4 pharmaceutics-16-00273-f004:**
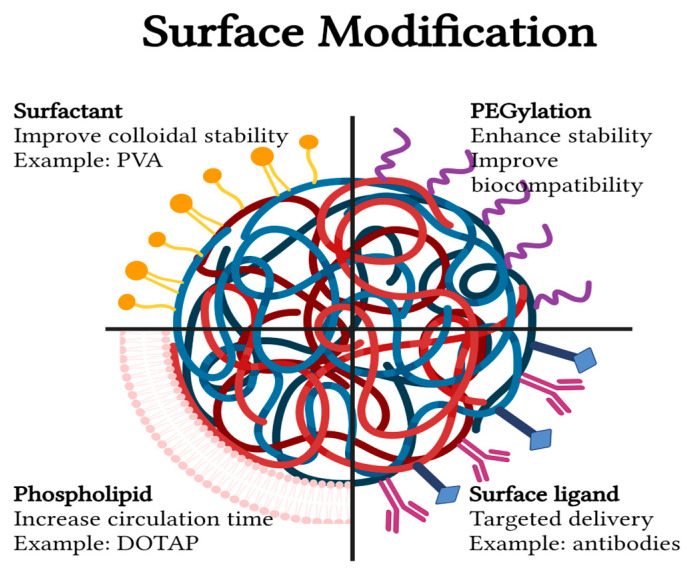
Surface modifications on PLGA NPs to enhance bone regeneration.

**Figure 5 pharmaceutics-16-00273-f005:**
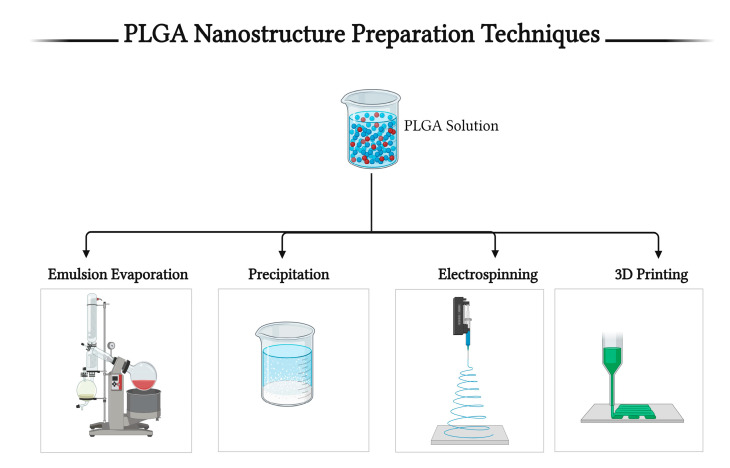
Different techniques for PLGA nanostructure preparation.

**Figure 6 pharmaceutics-16-00273-f006:**
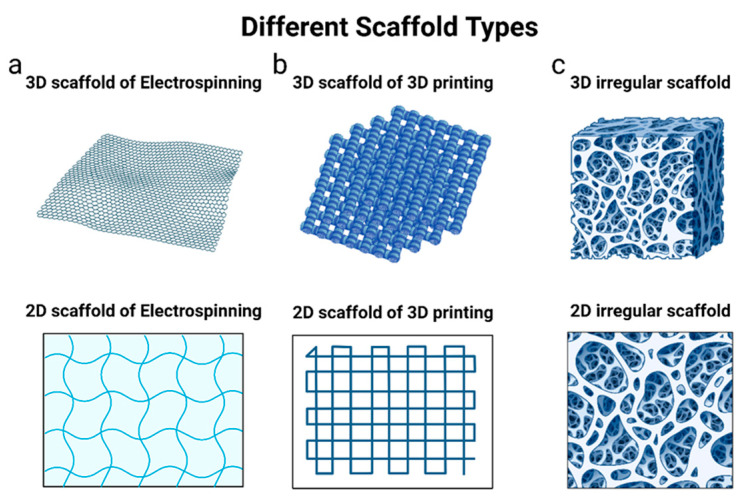
Different types of scaffolds according to preparation techniques: (**a**) electrospinning scaffolds, (**b**) 3D-printed scaffolds, and (**c**) irregular scaffolds.

**Figure 7 pharmaceutics-16-00273-f007:**
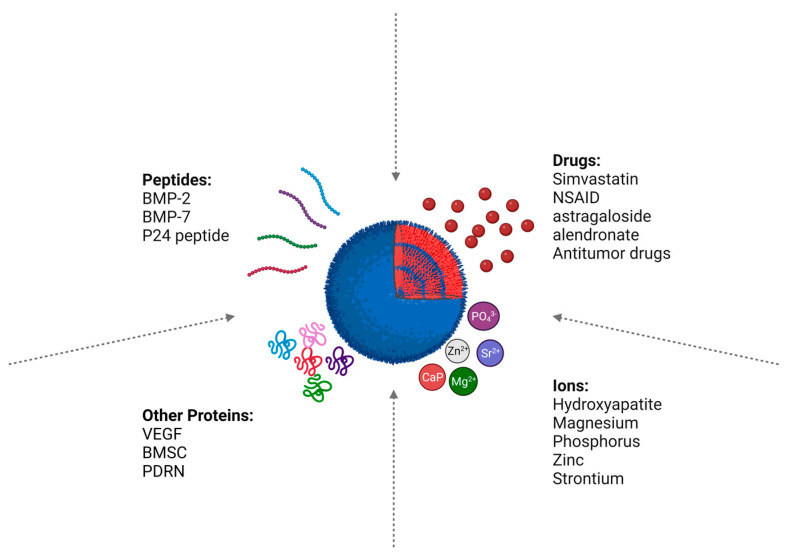
Bioactivated PLGA for enhanced bone regeneration.

**Table 1 pharmaceutics-16-00273-t001:** Nanoparticle’s classification based on composition.

Classifications	Advantages	Disadvantages	References
Organic nanoparticles: - Micelles - Dendrimers - Liposomes - Polymeric - Protein complexes	Hydrophilic, non-toxic, biodegradable, easy synthesis process, well-defined structure, changeable size, good surface characteristics, controlled drug delivery.	Sensitive to thermal and electromagnetic radiation such as heat and light, they are more susceptible to change in nature, leading to their elimination from the body.	[23,24,25]
Inorganic nanoparticles: - Silica - Quantum dots - Magnetic nanoparticles - Metal-based nanoparticles (silver, gold, copper, etc.) - Ceramics	- Stability, easy surface functionalization, high surface-to-volume ratio, ultraviolet-visible sensitivity, and electrical conductivity. - Catalytic, thermal, and antibacterial properties.	Long-term toxicity, genotoxicity, and oxidation vulnerability may induce an inflammatory response.	[26,27]
Carbon-based nanoparticles: - Fullerenes - Carbon nanotubes - Graphite, graphene, and graphene oxide - Nano-diamonds	Electrical conductivity, heat conductivity, good mechanical properties, high stability, high surface area, excellent optical activity.	- Can activate ROS-associated signaling pathways and trigger the release of cytokines. - The presence of bio-corona on the surface of carbon-based nanoparticles may alter their activity, biodistribution, pharmacokinetics, cellular uptake, toxicity, and clearance.	[28,29]

**Table 2 pharmaceutics-16-00273-t002:** PLGA nanosystems and their intended uses.

Intended Use	Active Agent	Targeted Delivery	PLGA Formulation	In-Vitro/ In-Vivo	Results	References
Anticancer	DOX	Use of FA for targeted delivery against folate receptors	PLGA/DOX/ γPGA/FA NPs	HeLa cells	PLGA/DOX/γPGA/FA NPs have targeted and pH-dependent release.	[101]
	DOX	Thermogel	DOX-loaded-liposome fabricated within PLGA-PEG-PLGA thermogel	4T1 cells (in vivo) BALB/c mice	The thermogel proved to have no burst, controlled DOX release in vitro, and enhanced anticancer activity in vitro and in vivo with fewer side effects.	[102]
	DOX	Anti-EGFR antibody cetuximab (C) Light-induced chemotherapy (NIR)	DOX/PLGA/PD/ PEG/C core-shell NPs	UMSCC 22A cells	The core-shell NPs with photothermal activity and targeting antibodies have enhanced and safer chemotherapeutic activity.	[103]
	DOX pEGFP DNA solution	FA	Polymeric-liposome-loaded-DOX/PLGA nanosphere complexed with pEGFP DNA	MDA-MB-231 cells	The core-shell nanospheres succeeded in co-delivery of DOX and pEGFP DNA into breast cancer cells.	[104]
	DOX	CPPs-LMWP [C24LMWP]	DOX/PLGA/C24-LMWP NP	A549/T, MCF-7/ADR, and 293T	LMWP delivered DOX/PLGA NPs by targeting MDR cancer cells overexpressing heparan sulfate proteoglycans.	[105]
	DOX (Adriamycin)		DOX/PLGA microspheres loaded HA/collagen scaffold (DOX/PLGA/HAC)	BMSC collected from the bone marrow of femurs of male Wistar rats (in vivo)	DOX/PLGA/HAC scaffolds exhibited bone repair activity with no obvious inflammatory signs, as well as enhanced antineoplastic activity.	[106]
	Diacetate acetyl curcumin (AC)		AC/PLGA/liposome	HeLa and HDFa cells	A new drug delivery system with theranostic applications.	[107]
	Mitoxantrone (MXT)	Ultrasound-responsive liposome	MXT/PLGA /Lip		Sustained release of NPs with ultrasound-responsive activity.	[108]
	Recombinant methioninase (rMETase)	Single-chain variable fragment (scFV) antibody	scFV/rMETase/PLGA/Lip	SGC-7901 cells	scFV/PLGA/Lip NPs have higher cellular uptake in gastric cancer cells. scFV/rMETase/PLGA/Lip enhanced the anticancer activity of rMETase.	[109]
	Cisplatin	Anti-VEGF antibody Avastin^®^	Avatin^®^/Lip/ PLGA/Cis	SiHa cells	PLGA forms stable Cis NPs with sustained release. Encapsulating the NP into Avastin^®^-conjugated liposomes enhances its intracellular uptake and thus its anticancer activity.	[110]
	Luteolin (L)	Antibody (PD-L1)	L/PD-L1/ PLGA/Lip	HepG2 cells	NPs with improved in vitro release profiles, cancer cellular uptake, and migration inhibition.	[111]
	Paclitaxel and elacridar (ELC)	Transferrin (Tf)	Tf/PTX-ELC /PLGA NPs	EMT6/AR1.0 cells	Co-delivery of PTX and P-gp inhibitors to overcome multidrug resistance and maintain intracellular therapeutic drug levels.	[112]
Antioxidant	Resveratrol		PLGA-oil nanohybrids (PONHs)/resveratrol	Normal monkey kidney (Vero) cells	PONH decreased cytotoxicity and improved the scavenging activity of resveratrol in vitro.	[113]
	Gallic acid (GA)		GA/PLGA	*S. aureus* HaCaT cells	GA/PLGA NPs with controlled release in vitro, excellent antioxidant activity, good antimicrobial activity against *S. aureus*, and good biocompatibility.	[114]
	Rutin (vitamin P) and NAAA inhibitor (URB894)		rutin/URB894/PLGA NPs	C-28 and NCTC-2544 cells	The co-encapsulation of rutin and URB894 in PLGA NPs resulted in synergistic antioxidant activity.	[115]
Antibiotic	Clindamycin		Clindamycin/ PLGA NPs		Formation of sustained clindamycin release up to 3 months.	[116]
	Gentamicin (gentAOT)	Zirconia scaffolds	gent AOT/ PLGA NPs	*S. aureus* osteoblast-like MG-63 cells	gent AOT/PLGA NPs adequately inhibited the growth of *S. aureus*.	[117]
Anti-atherosclerosis	Simvastatin (SIM)		SIM/PLGA/Lip	RAW 264.7 cells (in vivo) atherosclerotic model rabbits	SIM/PLGA/Lip NPs showed increased circulation time and enhanced athero-protective activity.	[118]
Anti-restenosis	Dexamethasone (DEX) or Rapamycin (Rap)		PEO-PLGA/DEX PEO-PLGA/Rap NPs were then coated with gelatin		In vitro-controlled release of coated NPs.	[119]

Abbreviations: FA (folic acid), PGA (poly(L-γ-glutamic acid), HeLa (human cervical cancer cell line), 4T1 (murine breast cancer cell line), PD (polydopamine), UMSCC 22A (Human head and neck squamous carcinoma cell line), CPP (cell-penetrating peptides), LMWP (low molecular weight protamine), A549/T (human lung cancer drug-resistant cell line), MCF-7/ADR (human breast cancer drug-resistant cell line) and 293T (human embryonic kidney transformed cell line), HDFa (human dermal fibroblast), and PD-L1 (programmed death ligand-1).

**Table 3 pharmaceutics-16-00273-t003:** Techniques of PLGA nanosystems preparation.

**Method**	**Advantage**	**Disadvantage**	**References**
Emulsion–solvent evaporation	Simple, spherical particles	Polydisperse particle sizes and high sheer forces degrade the active agent	[136,137,138]
Nanoprecipitation	High yield and reproducibility, and high encapsulation efficiency of hydrophobic drugs	Polydisperse particle sizes and high sheer forces degrade the active agent	[114,115,118,139]
Electrospinning	Easily forms uniformly fibrous, and multilayered scaffolds	Need electrospinning equipment and 2D nanofibrous membranes	[117,135,140]
3D printing	Adjustable sizes and shapes of the fabricated and monodispersed scaffolds	3D printer is required; it is not compatible with all types of polymers, and drugs may degrade during the drying step	[141,142,143]

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
