# Peer review of "Harnessing the Potential of PLGA Nanoparticles for Enhanced Bone Regeneration"

_pharmaceutics, 2024, doi:10.3390/pharmaceutics16020273_

Round 1

Reviewer 1 Report

Comments and Suggestions for Authors

The review is very well written: easy to read and exhaustive as for the field explored. It may be accepted in its current form for publication.

Author Response

Thank you sincerely for your thorough and positive review of our manuscript. We greatly appreciate your kind words. We are committed to maintaining the high standards of your journal, and we are grateful for your confidence in the readiness of our manuscript for publication. We look forward to the possibility of contributing to your esteemed journal.

Reviewer 2 Report

Comments and Suggestions for Authors

The paper is well-arranged and well-written. In my opinion, a minor revision is required before acceptance of this manuscript.

1.      In section 2, please cover other forms of nanostructured materials such as nanofiber, nanorods, nanowires, etc, and then explain the importance/advantages of nanoparticles in bone regeneration.

2.      In Table 3, since the electrospinning technique is used to prepare polymeric nanofibers, it should not be included here. Otherwise, the heading of section 5 should be changed to PLGA-based nanostructures preparation.

3.      Section 5.3. should be enriched with the recent literature. For example https://doi.org/10.3390/pharmaceutics11070305

Author Response

Reviewer 2 Comments:

The paper is well-arranged and well-written. In my opinion, a minor revision is required before acceptance of this manuscript.

We sincerely appreciate your guidance and the valuable input.  Your input has been instrumental in enhancing the overall merit of the paper.

In response to the given suggestions, we have successfully implemented certain modifications to the manuscript. These alterations have been conspicuously highlighted in yellow throughout the document. 

  1. In section 2, please cover other forms of nanostructured materials such as nanofiber, nanorods, nanowires, etc, and then explain the importance/advantages of nanoparticles in bone regeneration.

Answer: We would like to thank the reviewer for the useful comments and suggestions. The mentioned nanostructured forms were added to section 2 (line 82-92) followed by importance of nanoparticles in bone regeneration (line 96-100).  Added sections are highlighted.

  1. In Table 3, since the electrospinning technique is used to prepare polymeric nanofibers, it should not be included here. Otherwise, the heading of section 5 should be changed to PLGA-based nanostructures preparation.

Answer: We appreciate this suggestion. The heading of section 5 has been changed to PLGA based nanostructures preparation.

  1. Section 5.3. should be enriched with the recent literature. For example, https://doi.org/10.3390/pharmaceutics11070305.

Answer: Thank you for this useful suggestion. Additional literature has been incorporated to section 5.3 from the suggested paper  (https://doi.org/10.3390/pharmaceutics11070305), as well as other literature (line 498-501 and 506-523).

Reviewer 3 Report

Comments and Suggestions for Authors

The review article titled "Harnessing the Potential of PLGA Nanoparticles for Enhanced Bone Regeneration" by Hassan et al. presents a well-structured and informative manuscript, offering state-of-the-art advancements in the field of bone tissue regeneration. I recommend its publication.

However, a few minor comments are worth noting:

The introduction contains an abundance of foundational information, such as the BTE concept and an introduction to bone/cartilage tissue, which is commonly available in published review articles or textbooks. The authors should restructure the introduction to maintain logical coherence while emphasizing the primary focus of the manuscript.

There are instances of repeated abbreviation, such as PLGA, throughout the manuscript. It would be beneficial to cross-check and avoid redundant definitions.

The challenges associated with PLGA nanoparticles indeed include controlling drug release and achieving consistent size distribution. However, chemical modification and various nanoparticle preparation techniques can help address these challenges to some extent. Pursuing on-demand control over biodegradation and drug release presents an even more intriguing and challenging endeavor.

Author Response

Reviewer 3 Comments:

The review article titled "Harnessing the Potential of PLGA Nanoparticles for Enhanced Bone Regeneration" by Hassan et al. presents a well-structured and informative manuscript, offering state-of-the-art advancements in the field of bone tissue regeneration. I recommend its publication.

However, a few minor comments are worth noting:

The introduction contains an abundance of foundational information, such as the BTE concept and an introduction to bone/cartilage tissue, which is commonly available in published review articles or textbooks. The authors should restructure the introduction to maintain logical coherence while emphasizing the primary focus of the manuscript.

There are instances of repeated abbreviation, such as PLGA, throughout the manuscript. It would be beneficial to cross-check and avoid redundant definitions.

The challenges associated with PLGA nanoparticles indeed include controlling drug release and achieving consistent size distribution. However, chemical modification and various nanoparticle preparation techniques can help address these challenges to some extent. Pursuing on-demand control over biodegradation and drug release presents an even more intriguing and challenging endeavor.

Answer: We would like to express our gratitude for the thorough review of our manuscript. We are grateful for the constructive feedback and appreciate the time and effort that you have dedicated to evaluating our work.

Thank you for this suggestion, we totally agree with this comment. Some changes and some paragraphs have been omitted from the introduction part to make it more concise, changes have been made using MS Word tracking function. The abbreviations were also rechecked throughout the manuscript.

Reviewer 4 Report

Comments and Suggestions for Authors

The study by Hassan et al. (Harnessing the Potential of PLGA Nanoparticles for Enhanced Bone Regeneration) reviews the advances in the employment of poly(lactic-co-glycolic acid) (PLGA) as bone regeneration material. In the manuscript, the authors provide some insights into the properties, modifications, and fabrication of PLGA nanoparticles for bone tissue regeneration.

I think the report represents valuable information for this research field. However, considering the length of the text, the number of figures seems to be inadequate. I think the authors must provide figures from some striking works in the literature. This would help in catching the attention of the reader.  For each section, at least one figure is required.  

Comments on the Quality of English Language

Minor editing of English language required

Author Response

Reviewer 4

The study by Hassan et al. (Harnessing the Potential of PLGA Nanoparticles for Enhanced Bone Regeneration) reviews the advances in the employment of poly(lactic-co-glycolic acid) (PLGA) as bone regeneration material. In the manuscript, the authors provide some insights into the properties, modifications, and fabrication of PLGA nanoparticles for bone tissue regeneration.

I think the report represents valuable information for this research field. However, considering the length of the text, the number of figures seems to be inadequate. I think the authors must provide figures from some striking works in the literature. This would help in catching the attention of the reader.  For each section, at least one figure is required.  

Answer: Thank you for your thoughtful comment, we sincerely appreciate your feedback. In response to your suggestion, we have incorporated additional figures into the manuscript (Figure 2, 4, 5 and 7) to enhance the clarity and support the contents of the review.  

Round 2

Reviewer 4 Report

Comments and Suggestions for Authors

I recommend the publication of the report. 

Comments on the Quality of English Language

Minor editing of English language required